# Tri-Response Police, Ambulance, Mental Health Crisis Models in Reducing Involuntary Detentions of Mentally Ill People: Protocol for a Systematic Review

**DOI:** 10.3390/ijerph18158230

**Published:** 2021-08-03

**Authors:** Julia Heffernan, Ewan McDonald, Elizabeth Hughes, Richard Gray

**Affiliations:** 1School of Nursing and Midwifery, La Trobe University, Bundoora, Melbourne, VIC 3086, Australia; Ewan.McDonald@latrobe.edu.au (E.M.); R.Gray@latrobe.edu.au (R.G.); 2Faculty of Medicine and Health, University of Leeds, Leeds LS2 9JT, UK; E.C.Hughes@leeds.ac.uk

**Keywords:** mental health, mental illness, police, ambulance, pacer, psychosis, self-harm, involuntary detention, section, assessment, systematic review, protocol

## Abstract

Police, ambulance and mental health tri-response services are a relatively new model of responding to people experiencing mental health crisis in the community, but limited evidence exists examining their efficacy. To date there have been no systematic reviews that have examined the association between the tri-response model and rates of involuntary detentions. A systematic review examining co-response models demonstrated possible reduction in involuntary detention, however, recommended further research. The aim of this protocol is to describe how we will systematically review the evidence base around the relationship of the police, ambulance mental health tri-response models in reducing involuntary detentions. We will search health, policing and grey literature databases and include clinical evaluations of any design. Risk of bias will be determined using the Effective Public Health Practice Project Quality Assessment Tool and a narrative synthesis will be undertaken to synthesis key themes. Risk of bias and extracted data will be summarized in tables and results synthesis tabulated to identify patterns within the included studies. The findings will inform future research into the effectiveness of tri-response police, ambulance, and mental health models in reducing involuntary detentions.

## 1. Introduction

Involuntary detention is a common mechanism used to compel people who appear acutely mentally ill for a mandatory psychiatric assessment or period of observation [1,2,3,4,5]. It is a controversial power provided in mental health legislation, generally to doctors, mental health workers, police officers and in some areas, paramedics. Involuntarily detaining a person brings with it additional powers which allow force to be used upon the person to complete that detention as included in mental health legislation [5,6]. Such powers may include forcing entry into the persons property, searching their person and property, using physical force and restraint, and the use of chemical sedation [1,2,3,4,5].

Mental health consumers and carers report that involuntary detention is a traumatic, humiliating and often frightening experience, particularly when involving police or law enforcement agencies, which negatively impacts their overall mental wellbeing [1,2,3,4,5,6,7,8,9,10,11]. It is consistently demonstrated that involuntary detention invokes loss of perceived independence, worsening of paranoid beliefs, terror and distress, re-traumatization, and powerlessness, particularly for those who experienced restrictive practices such as restraint and forcible giving of medications [1,2,3,4,5,6,7,8,9,10,11].

Police will often invoke involuntary detention in the absence of other mechanisms to ensure prompt assessment of a person experiencing mental illness [2,12,13]. Similarly, ambulance paramedics frequently respond to mental health crisis, often co-responding with police [14,15]. High rates of involuntary detentions have significant resourcing impacts on emergency services and particularly emergency departments (EDs) and hospitals who are required to undertake the assessments. This includes higher numbers of patients in EDs, the need for greater supervision, prevention of absconding, and pressures related to involuntary detention assessment times [16,17,18,19].

Research suggests significant increases in the rates of involuntary detentions [13,16,17,18,19,20]. An analysis of involuntary detention rates in the United States determined that the rate of involuntary detentions in 22 states was increasing by 13% every year between 2012 to 2016 [17]. As a result, health services, police and ambulance services have observed the need to provide mental health expertise directly into police and ambulance presentations involving mental health patients.

A trained mental health clinician can provide expert assessment in the field, negating the need to invoke involuntary detention or transporting the person to hospital for assessment [3,21]. A number of models are being trialled across a number of developed countries including adding mental health workers into police or ambulance call centers, co-response mobile crisis services which may team a mental health worker with a paramedic or police officer, or the tri-response model which incorporates all three agencies [21]. Police, Ambulance, Clinician Early Response (PACER) is a tri-response mobile service which teams a mental health clinician, police officer and ambulance paramedic together in a first responder vehicle to attend mental health crisis in the community and is one of a number of models being trialled to meet this need, yet it requires further exploration to assess its efficacy in reducing involuntary detentions. Currently, tri-response models are operating in parts of Australia, the United Kingdom, the United States and Europe.

A systematic review of police mental health co-responder models was undertaken in 2018 to identify and describe the different models, identify the types of service users who came in contact with the models and to evaluate their effectiveness [21]. The authors included 26 papers into the review and concluded that the co-responder police mental health models may reduce rates of involuntary detention of mentally ill people [21]. However, they opined that further research was required, and at the time of writing, no review has been conducted to evaluate the tri-service model [21].

### Research Objectives

This protocol describes the objective of the future review which is to synthesize the available evidence regarding the effects of the PACER mobile crisis service and similar tri-response models in diverting patients from hospital and reducing unnecessary involuntary detention.
To clarify the evidence base around the relationship of the tri-response model in reducing involuntary detentions of people experiencing mental health crisis. This systematic review will clarify the available research through all available studies published in journals and abstracts, which meet inclusion criteria.To compare the rate of tri-response involuntary detentions which convert to hospitalizations with involuntary detentions made by police and/or ambulance paramedics.To compare the rate of involuntary detentions by the tri-response model with rates of involuntary detentions made by police and/or ambulance paramedics.

## 2. Materials and Methods

### 2.1. Review Question

The aim of the systematic review is to determine the safety and effectiveness of the tri-response crisis model compared with routine intervention in people experiencing a psychiatric crisis. This protocol sets out the following review design.

### 2.2. Design

We will include clinical studies of any design which answers the research question and exclude qualitative studies. Our protocol complied with the Preferred Reporting Items for Systematic Reviews and Meta-Analyses Protocols checklist 2015 (PRISMA-P) [11].

Our review is registered with OSF: doe:10.17605/OSF.IO/3EMRV.

### 2.3. Eligibility Criteria

Participants are experiencing an acute mental health crisis which has precipitated an emergency response or PACER response. Acute as defined for the purpose of this systematic review, pertains to a mental health presentation which is severe or intense in nature, and poses a risk to the person or others because of mental illness. It refers to the need for immediate assessment for which an emergency response is appropriate. Examples of such presentations include florid psychosis, mania, suicidal behaviour, and behavioural disturbance. Presentations can be unknown to the responders and requiring crisis assessment. Such presentations may include a person behaving in a confused and agitated manner in a public place.

We will include studies meeting the following inclusion criteria:Patients meeting the participant criteria;The exposure or intervention is tri-response police, ambulance, mental health models;All participant demographics (age, gender, ethnicity, etc.);All countries of publication;Published in English.

### 2.4. Exclusion Criteria

The focus of the review described in this protocol is to synthesis the evidence relating to the tri-response PACER model; therefore, co-response models will be excluded. Co-response models include police and mental health clinician; paramedic and mental health clinician; non-mobile models, mental health clinicians embedded within police station or emergency communications centers. For example, some known services are titled “PACER” but stand for “Police and Clinician Early Response” and do not include an ambulance paramedic. Others are termed “psychiatric ambulances” and refer to a model where a clinician is embedded with a paramedic team in an ambulance vehicle. It is considered that the co-response model is not representative of the tri-response model.

### 2.5. Searching the Grey Literature

Grey literature will be used in the review and is expected to include service evaluations of tri-response PACER models, relevant theses or dissertations, research and committee reports, and government reports. Reference lists of relevant studies, citation searching and searching relevant internet resources will also be included

### 2.6. Comparator Intentions

Any comparator will be included in the review however the primary comparator is a standard emergency service response, that refers to a patient experiencing mental health crisis who is reviewed by an ambulance or police team, as part of standard emergency service response.

### 2.7. Search Strategy

The Medline search strategy can be found in Table 1. This review will incorporate both automated and manual searches. Literature relating to tri response PACER models will most likely come from health databases, however police databases will also be searched including:

Health databases:CINAHL Complete;Health and Medical Complete (ProQuest);Medline (OVID);Joanna Briggs Institute EBP Database;Health Collection (Informit);PsychINFO;PsycARTICLES;ANZCTR.

Policing databases:ProQuest Criminal Justice Database;Australian Federal Police Digest;CINCH Australian Criminology Database.

Gray literature databases:Open Gray;Gray Source;ProQuest;Google.

To maximize outcomes of the literature search, we will ensure that synonyms are utilised as part of the search strategy. Several synonyms exist in this field of study and relate to the description of mental health patients, terminology used to describe involuntary detentions and the title of PACER. For example, “mental health” is a field of psychiatry and the term is interchangeable with “mental illness”. Involuntary detentions are commonly referred to as “sections” with specific countries and states referencing the individual section within their own legislation. Therefore, Boolean operators, plurals, truncations, and wildcards will be included in the search terms to maximize the search results and reduce researcher burden.

Furthermore, “PACER” is a commonly used term in the medical field of cardiology and will need to be paired with a cardiac search term exclusion. It is likely that the search will produce many co-response model studies despite strict search terms and will then need to undergo further screening to identify those which refer to the PACER tri-response. Finally, the search will need to filter out the clinician, given this is a highly changeable term which is unlikely to appear in many titles or abstracts and may impact the outcome of the search.

### 2.8. Data Retrieval

The selected studies will be exported from bibliographic databases to reference management software (Endnote)

### 2.9. Data Screening

Duplicate references will be removed in Endnote and citations exported into Covidence, a web-based software platform for systematic reviews including citation screening, review of full text articles, risk of bias assessment, extraction of study characteristics and outcomes, and exportation of data.

We will use two reviewers to review citations for inclusion in the review, initially using title and abstracts to screen against inclusion/exclusion criteria. Using this double screening approach which offers the following advantages: assurance that the inclusion/exclusion criteria are consistently applied, identification of and correction of mistakes, and avoidance of systematic errors [22].

Conflict will be resolved through discussion and any issues that are unable to be resolved will be referred to a third reviewer for resolution. This screening process aims to avoid inclusion of evidence with a subsequent risk of bias that could endanger the validity of the conclusions drawn in the review. These requirements aim to avoid the non-detection of relevant evidence with a subsequent risk of bias that endangers the validity of conclusions drawn from the evidence available. The relevant publications are selected in several steps. Following the first screening phase, full text screening will occur using the same process of double screening against the inclusion/exclusion criteria.

### 2.10. Data Extraction

Data will be extracted from papers using an extraction tool developed for this review using the review question as a guide. The data extracted will include details about the intervention, patient population, study methods, and outcomes of significance to the review question and objectives.

### 2.11. Risk of Bias in Individual Studies

Quality assessment of the included literature will be carried out at the point of data extraction. The Effective Public Health Practice Project Quality Assessment Tool (EPHPP) (https://merst.ca/wp-content/uploads/2018/02/quality-assessment-tool_2010.pdf, accessed on 18 April 2021). The EPHPP was developed in Canada by the Effective Public Health Practice Project and is an effective tool for evaluating a number of different study designs including Randomised Controlled Trials (RCTs), before and after intervention studies and case–control studies [23].

The tool has been assessed as having content and construct validity and measures six key domains: (1) selection bias; (2) study design; (3) confounders; (4) blinding; (5) method of data collection; and (6) withdrawals and dropouts, and two further components of intervention integrity and analyses [13]. The EPHPP was assessed for content validity using an iterative process of systematic repetition of sequences of testing different data [24]. The tool was reviewed for consistency of coding, interpretation, and examination of evidence tables using an expert group. The authors further assessed the validation process through evaluating the content of the tool and the individual categories for clarity, completeness, and relevance, as well and comparing the EPHPP with similar tools [24]. Furthermore, Test–retest reliability of the EPHPP was calculated over two occasions using two reviewers and a random selection of studies, with agreement between the two reviewers (Kappa 0.74: Kappa 0:61) [24].

Overall, the EPHPP has a strong methodological rating based upon its validity and reliability [24]. For this review, two reviewers will use the EPHPP to rate the study quality and a comparison of individual ratings will reach a consensus on each component. In the event of a lack of consensus, a third reviewer will apply the EPHPP to the contested study. The overall study quality will be rated based upon the combined component ratings using the following:

Strong—Four strong ratings with no weak ratings;

Moderate—Less than four strong rating and one weak rating;

Weak—Two or more weak ratings.

### 2.12. Data Synthesis

Narrative synthesis will be performed due to the inclusion of all study designs in the review which may yield literature that is not adequately clinically homogenous to allow for meta-analysis [25,26,27,28].

The synthesis will focus upon the intervention implementation and effect and grouped into themes. The synthesis will be undertaken and carried out using the following framework:Developing a theory of how the intervention works, why and for whom;Developing a preliminary synthesis of findings and included studies;Exploring relationships within and between studies;Assessing the robustness of the synthesis [29].

A preliminary synthesis of the findings of the studies and grey literature will be completed using the Cochrane Narrative Synthesis Advice [30]. Results will be tabulated to identify the patterns within the studies including population size, length of study, study design and outcomes and then transcribed into both descriptive and statistical format.

### 2.13. Meta Bias(es)

Bias refers to systematic error that skews the results of a study in a particular direction and leads to the acceptance of outcomes or results without considering the likelihood of unfair or misleading presentation [31,32,33]. There are many types of bias that can impact the findings of a systematic review. Publication bias refers to the inclusion of studies which are statistically significant or demonstrate favourable results but minimize the inclusion of studies which do not [31,32,33]. Studies which demonstrate efficacy, success, or confirm a researcher’s hypothesis, are more likely to be published and can threaten the validity of the review, providing an unbalanced summary of the evidence [33]. Publication bias can be minimized by the inclusion of grey literature in the review [31,32,33]. Removing grey literature from meta-analysis results in 15% larger estimates of treatment effects, less precise effect-size estimates and more significant results and using only published trials may result in greater treatment effect [33] (p. 234).

Similarly, selective reporting or selection bias occurs when research results are deliberately reported inaccurately in order to suppress or exclude negative or undesirable findings [31,32] This may result in findings which are skewed and unable to be reproduced in further study. The impact of selection bias can vary and may be influenced by the selector’s knowledge of the subject, existing collaborations, and informed opinion about the topic [31,32]. Where available, we will assess study protocols of literature to be included in the review and determine the completeness of the reporting. Selection bias can further be reduced by the inclusion of grey literature [33].

Inclusion of grey literature alone does not entirely alleviate the risk of publication bias or selective reporting. To minimize systematic error of the above biases, we will use the funnel plot measure of symmetry of the study effect in the included literature [33].

## 3. Results

Risk of bias and extracted data will be summarized and emphasized through tables, graphs and other diagrams used to compare the data [26]. Where possible point estimates (the value that represents a best estimate of effects) and interval estimates (an estimated range of effect, presented as a 95% confidence interval) will be presented in the results [26].

## 4. Discussion

This paper sets out the protocol for a planned systematic review and narrative synthesis. Each step has been carefully considered and planned to meet best practice standards for systematic reviews of health research.

## 5. Conclusions

There is considerable interest from consumer and carer groups to reduce the use of involuntary detention and restrictive practices of people experiencing mental illness, and to promote recovery and least restrictive care. This review will add to the body of evidence about tri-response mobile crisis response teams in responding to people in mental health crisis, and their capacity to reduce involuntary detentions.

## 6. Relevance to Clinical Practice

There is considerable interest from consumer and carer groups to reduce the use of involuntary detention and restrictive practices of people experiencing mental illness, and to promote recovery and least restrictive care. This review will add to the body of evidence about cross-agency mobile crisis response teams in responding to people in mental health crisis, and their capacity to reduce involuntary detentions.

## Figures and Tables

**Table 1 ijerph-18-08230-t001:** Medline Search.

Search ID	Search Term	PICO Element
S1	Mental Health	Population
S2	Psychiatr*	
S3	Mental* Ill*	
S4	Crisis	
S5	Emergency	
S6	Acute*	
S7	Urgent	
S8	Relapse	
S9	Self-harm*	
S10	Self-injur*	
S11	Psycho*	
S12	Mani*	
S13	Suicid*	
S14	S1 or S2 or S3 and S4 or S5 or S6 or S7 or S8	
S15	S14 and S9 or S10 or S11 or S12 or S13	
S16	PACER*	Intervention
S17	Police	
S18	Ambulance	
S19	Nurse	
S20	Paramedic	
S21	Law Enforcement	
S22	Street Triage	
S23	Joint	
S24	Mobile	
S25	Tri-response	
S26	S16 or S17 or S18 or S19 or S20 or S21	
S27	S22 or S23 or 24 or S25	
S28	Emergency Service* or S26	Comparator
S29	Hospital*	Outcome
S30	Section*	
S31	Involuntary	
S32	Detention	
S33	Section 136	
S34	Legislation	
S34	S29 or S30 or S31 or S32 or s33 OR s34

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
