# Peer review of "Tri-Response Police, Ambulance, Mental Health Crisis Models in Reducing Involuntary Detentions of Mentally Ill People: Protocol for a Systematic Review"

_ijerph, 2021, doi:10.3390/ijerph18158230_

Round 1

Reviewer 1 Report

Thank you for the opportunity to review this paper. It deals with a subject of high social relevance. However, I do believe that there are some flaws related to design and construct validity that must be addressed.

The cooperation of Paramedics and Police is regulated nationally, it's worth mentioning that fact. Seems to me, that the Authors concentrate mostly on their own country.
As mentioned before, PACER is not used in all countries- the Authors could add information, where it is used. Maybe more details on the model would be helpful?

Adding the diagram with an explanation of the review process would be helpful.

Why do the Authors decide to use grey literature only? Which database have the Authors searched?

Lack of dot line 129.

Reviewer 2 Report

The paper focuses on the demonstration of a systematic review protocol on a topic of interest: crisis intervention with people with mental disorders. However, its contribution is limited to this demonstration, which is quite disappointing from a knowledge advancement point of view. I recommend that the authors review the objectives and the research question in order to better describe what the paper presents. It would also be interesting to have a better argument on the need to evaluate the PACER model vs the co-response models. For example, why it is believed that the PACER model could be more efficient than the other models. Moreover, given the nature of the paper, it would have been more appropriate for the introduction to address the methodological aspects of systematic reviews. 

Reviewer 3 Report

The work presented is of enormous interest and relevance.
It is worth paying attention to such a controversial topic as involuntary detentions.
The following are a number of issues that could improve the work presented: 
- The objective/s should appear at the end of the introduction.
- The authors should specify the search string in more detail.

Reviewer 4 Report

A carefully thought out and well written paper on an important topic.

The authors have taken a great deal of care to ensure that any selection bias in their review is minimised as far as possible by the use of solid methodology. The paper is set out logically and persuades me that worthwhile results will be achieved from the review. Guidance on minimisation of involuntary admissions is a worthwhile goal, particularly at a time when the rate of compulsory treatment seem to be at a high level in some jurisdictions (Reference 3) and when limited evidence on its efficacy is available.

A few minor issues:

'Trialed' should be 'trialled'.

2.8, para 2. Correct sentence starting "We will ensure that ..."

2.12 last para. "4 strong ratings" should be "Four strong ratings"

2.14, para 1. Sentence starting "Publication bias ..." is awkwardly expressed.

2.14, para 2. Correct "This see's results...."   

Round 2

Reviewer 1 Report

I accept all changes with one request- if the Authors could be more specified in counties that use the PACER model.

Author Response

We have addressed the request in the first re-write of the manuscript, tracked changes in lines 60, 68 and 69

Reviewer 2 Report

Again, the objectives of the article are not aligned with what is actually presented in the paper. I recommend that the authors specifically mention that the article aims to present a protocol with results to come. It's clearer in the summary, but not in the paper. The objectives could be presented for discussion to announce the next paper.

Author Response

We consider that we have explicitly stated that this paper is a protocol. It is written in the subheading, the title of the paper, the abstract and throughout the document.